# Impact of short-term change of adiposity on risk of high blood pressure in children: Results from a follow-up study in China

Yi-de Yang[ORCID][1,2☯]*, Ming Xie[1☯], Yuan Zeng[1☯], Shuqian Yuan[1], Haokai Tang[1], Yanhui Dong[2], Zhiyong Zou[2], Bin Dong[2]*, Zhenghe Wang[3], Xiangli Ye[1], Xiuqin Hong[1], Qiu Xiao[4], Jun Ma[2]

**1** Key Laboratory of Molecular Epidemiology of Hunan Province, School of Medicine, Hunan Normal University, Changsha, China, **2** Institute of Child and Adolescent Health, School of Public Health, Peking University Health Science Center, Beijing, China, **3** Department of Epidemiology, School of Public Health, Southern Medical University, Guangzhou, Guangdong, China, **4** College of Information Science and Engineering, Hunan Normal University, Changsha, China

☯ These authors contributed equally to this work.
* yangyide@hunnu.edu.cn (YY); bindong@bjmu.edu.cn (BD)

**Data Availability Statement:** The datasets generated and/or analyzed during the present study are not publicly available, since ethics approval and participants' consent does not allow public sharing

## Abstract

This study aimed to examine the impact of short-term adiposity change on risk of high blood pressure (HBP), and to assess the low limit range of body mass index (BMI) and waist-to-height ratio (WHtR) reduction proposed to decrease the HBP risk in children. Children were longitudinally surveyed at baseline and after a short-term follow-up. General obesity (GOB) is categorized by age and gender-specific BMI cut-off points, abdominal obesity (AOB) by WHtR. Logistic regression model was used to estimate relations between adiposity change and HBP risk with adjustment of covariates. A total of 28,288 children (median of baseline age:10 years) were involved with follow-up of 6.88±1.20 months. After the follow-up, 9.4% of the children had persistent general obesity (GOB), 2.8% converted from GOB to non-GOB, 0.9% had newly developed GOB. When compared with children remained non-GOB, children with continuous GOB status, newly developed GOB, converting from GOB to non-GOB had 5.03-fold (95%CI: 4.32~5.86), 3.35-fold (95%CI: 1.99~5.65), 2.72-fold (2.03~3.63) HBP risk, respectively. Similar findings were observed for abdominal obesity (AOB). Reduction of 0.21–0.88 kg/m² of baseline BMI (0.86–3.59%) or 0.009–0.024 of baseline WHtR (1.66–4.42%) in GOB or AOB children, respectively, was associated with significant decrease in HBP risk. Children with persistent obesity, newly developed obesity, or converting from obese to non-obese had significantly higher HBP risk. For children with GOB or AOB, reduction of <3.6% in BMI or <4.5% in WHtR could decrease the HBP risk.

## Introduction

The prevalence and magnitude of pediatric high blood pressure (HBP) increased dramatically in recent years, which poses a heavy burden for cardiovascular diseases (CVDs) prevention [1]. As the most important cause of CVDs, HBP composed the largest number of attributable disability-adjusted life-years (DALYs) in China [2].

of data, only available from the author (Email: yangyide2007@126.com or bindong@bjmu.edu. cn) and also the Medical Ethical Committee of the Peking University Health Science (Email: llwyh@bjmu.edu.cn) upon reasonable request.

**Funding:** This research was funded by the National Natural Science Foundation of China (81903336), Hunan Province College Students Research Learning and Innovative Experiment Project (S202110542057)the Hunan Provincial Natural Science Foundation of China (2019JJ50376 & 2020JJ5386), Scientific Research Project of Hunan Provincial Health Commission (202112031516), Key Project of Hunan Provincial Science and Technology Innovation (2020SK1015-3) and a Project Supported by Scientific Research Fund of Hunan Provincial Education Department (18A0028). The funders had no role in the study design, data collection, data analysis, writing of this article or interpreting the results.

**Competing interests:** The authors have declared that no competing interests exist.

**Abbreviations:** AOB, abdominal obesity; BMI, Body mass index; BP, blood pressure; DBP, diastolic blood pressure; GOB, general obesity; HBP, high blood pressure; SBP, systolic blood pressure; SSB, sugar-sweetened beverage; WHtR, waist-to-height ratio.

The sharp increase of overweight and obesity prevalence contributes greatly to the epidemic of HBP among children [3]. The association between obesity and blood pressure (BP) levels or hypertension has been demonstrated in numerous cross-sectional studies [4–6]. Also, intervention studies against obesity showed that weight management will additionally benefit BP profile [7]. But in a natural life course, it remains unclear whether the change of general or abdominal fat accumulation status without intervention would impact the risk of HBP in childhood.

In addition, a national secular trend study of BP and obesity among Chinese children showed that the magnitude of impact of obesity on HBP increased sharply in recent years [1]. Therefore, in a natural life course the benefit of conversion of obesity to non-obesity and the risk of newly developed obesity or persistent obesity during childhood may also vary over time. Accordingly, we made a hypothesis that the conversion of obesity to non-obesity could be beneficial for reducing the risk of HBP, and the newly developed or persistent obesity is detrimental to the BP profile among childhood.

A previous weight management study among overweight children show that the reduction of just 5% of body mass index (BMI) is associated with improvements in lipid profiles and insulin sensitivity in six months [8], and a meta-analysis of lifestyle intervention studies showed that BMI standard deviation score (SDS) reduction of >1 SDS is associated with better BP profiles [7]. However, in a natural life course without intervention, for obese children the minimum range of adiposity reduction proposed to reduce the HBP risk is not known now. Hence, our study aimed to identify how the general obesity (GOB) or abdominal obesity (AOB) status will change in a short-term follow up, and to demonstrate the relationship between different obesity status change and BP profiles, including BP levels and risk of HBP in children, and also assess the low limit range for reduction of adiposity parameters (such as BMI and waist-to-height ratio) in a natural growth process proposed to be beneficial for reducing HBP risk in obese children.

## Subjects and methods

### Study design and participants

Data were deprived from a multi-center school-based obesity intervention program (https://www.clinicaltrials.gov/, Registration number: NCT02343588) which involved more than 70,000 children aged 6–17 years based on a random sampling method from 7 provinces (including Hunan, Liaoning, Ningxia, Guangdong, Shanghai, Chongqing, and Tianjin) in China at autumn of 2013. A detailed description of the program was published before [9, 10]. For the enrollment of subjects, the inclusion criteria were as follows: grade of 1 to 5 in primary schools and grade of 6 to 8 and 10 to 11 in secondary schools. And the exclusion criteria were as follows: Suffering or any history of disabilities, cardiovascular diseases, asthma, etc. For students in grade 1 to 5 in primary schools, the age of them was supposed to be 7–11 years old. But actually, in the real life, some children aged 6 years old also started primary school. Therefore, the age range in the registration trial (7–18 years old) is slightly different that of the present study (6–18 years old), also the age range of our paper is same with another paper about the effectiveness of this lifestyle intervention study. The length of follow up was 6.88 months (mean value), which is shorter than the length of follow up described in the study registry (9 months). This is because our study was conducted in 7 provinces (a total of 94 schools), the length of baseline physical examination and survey was almost a month, and the follow-up physical examination and survey also take a month to complete, so the range of the follow-up is set at 9 months, but for every participant at different schools the length of follow-up might be slightly different, and the mean value of follow-up is 6.88 months. To study the natural

change of obesity status and its associations with risk of HBP, we only included participants of the control group of this intervention program. A total of 28,288 participants without any intervention were involved in the present study, with complete data of baseline and the follow up anthropometric data (height, weight, waist circumference and blood pressure). This study was approved by the Medical Ethical Committee of the Peking University Health Science Center (IRB0000105213034). Written informed consent was obtained from all children and their parents.

## Measurements and definitions

Anthropometric measurements, such as height, weight, waist circumference and blood pressure were measured according to standardized protocols. Height was measured to the nearest 0.1cm with shoes. Weight was measured to the nearest 0.1 kg by a Lever type weight scale. Every indicator was measured twice, and the mean value of the two measurements were used for final analyses. BMI (kg/m$^2$) was calculated as weight(kg) divided by height(m) squared. Participants were categorized as generally obese if the BMI was larger than the age-and gender-specific cut-off points, according to the standard "BMI Reference for Screening for overweight and obesity among school-age children and adolescents" published by National Health Commission of the People's Republic of China [11] (S1 Table). Waist circumference was measured to the nearest 0.1cm, at the midway between the superior border of participant's iliac crest and the lowest rib with a flexible nylon tape. Waist-to-height ratio (WHtR) was calculated as waist circumference (cm) divided by height (cm). Participants were categorized as abdominally obese if their WHtR was not less than 0.5 [12, 13].

BP was measured by a mercury sphygmomanometer and it was measured two times, with at least five-minute interval between each measurement. If the measured difference was >10 mmHg, measurement was repeated until the final two measures differed <10 mmHg, and the mean of the final two measures was used in analyses. Children were asked to remain quiet and to sit still while each reading was being taken. Systolic blood pressure (SBP) was defined as the onset of "tapping" Korotkoff sound (K1), and diastolic blood pressure (DBP) was defined as the fifth Korotkoff sound (K5). High BP is defined as SBP and/or DBP ≥ the age-, gender- and height-specific 95th percentile [14].

## Covariates

Data of demographic information (birth date, gender and area), fresh fruit and vegetable consumption, sugar-sweetened beverage (SSB) intake, and time of vigorous or moderate physical activities at baseline were collected by a standard questionnaire [15, 16] (S1 File). Area was categorized as rural area and urban area of the school location. Frequency (days per week) and amount (servings) of fruits and vegetables, separately, consumed in the past week were investigated. The average of daily intake of vegetables was calculated as follows: average daily fresh vegetable consumption = [frequency × (servings of vegetable consumed in each of these days)]/7 [16], and the size of one serving of fruit or vegetable is as the size of a fist of ordinary adult (S1 Fig). For the average daily fresh fruit consumption, a similar equation was used as well. For SSB intake, every child answered two questions of the frequency of the intake and usual portion size(cups) of all SSB, including orange juice, Coca-Cola, Nutrition Express, Sprite and Red Bull (one cup = 250 ml). We combined these items to calculate the weekly SSB intake [17]. Physical activities included vigorous physical activities (activities which obviously increase the breathing and heart rate, for example, running, football, basketball et al.), and moderate physical activities (activities which increase the breathing and heart rate to some extent, for example bicycle, table tennis, calisthenics and badminton et al.). Frequency (days

per week) and time (specific hours and minutes for each of these days) spent on these activities in the past week were investigated. We used the equation: average daily time = [days × (time in each of those days)]/7, to calculate specific average time each day the children participated in vigorous physical activities or moderate physical activities [1].

## Statistical analysis

Mean/Medians with standard deviation (SD) / inter quartile range (IQR) were used to describe numerical variables. Frequencies were used to describe the categorical variables. Chi-square tests were used to compare the categorical variables between different groups, independent t tests were conducted quantitative variables between two groups, one-way ANOVA were used to compare quantitative variables in multiple groups. Adjusted means and 95% confidence interval of BP levels and BP changes were estimated by multivariate general linear model with adjusting gender, age, BMI, area, fruit consumption, vegetable consumption, SSB intake, and physical activity. Multivariable logistic regression model was used to examine the association between different obesity status change and the risk of HBP during six-month follow up adjusted for potential baseline covariates.

For children with GOB or AOB at baseline, we categorized their adiposity parameter (BMI or WHtR) change during the follow-up as quintiles (Quintile 1: $\leq P_{20}$, Quintile 2: $P_{20}\sim P_{40}$, Quintile 3: $P_{40}\sim P_{60}$, Quintile 4: $P_{60}\sim P_{80}$, Quintile 5: $>P_{80}$), and compared the risk of HBP in different quintiles with multivariate logistic models.

We additionally conducted a series of sensitivity analyses to test the robustness of the results: (1) We added height change from baseline to follow up survey in the models to examine the growth effect on the associations between obesity status change and risk of HBP. (2) We included the baseline HBP status as a covariate in the models to investigate the influence on the associations. (3) For children with GOB or AOB at baseline, we also categorized their adiposity parameter (BMI or WHtR) change during the follow-up as ten equal groups (Group 1: $\leq P_{10}$, Group 2: $P_{10}\sim P_{20}$, Group 3: $P_{20}\sim P_{30}$, Group 4: $P_{30}\sim P_{40}$, Group 5: $P_{40}\sim P_{50}$, Group 6: $P_{50}\sim P_{60}$, Group 7: $P_{60}\sim P_{70}$, Group 8: $P_{70}\sim P_{80}$, Group 9: $P_{80}\sim P_{90}$ and Group 10: $>P_{90}$) and compared the risk of HBP in different groups with multivariate logistic models. (4) We used an international definition of child obesity developed by the International Obesity Task Force (IOTF) and the World Health Organization (WHO) standard of obesity for 5–19 years old children to test the associations [18–24]. (5) We also used BMI standard deviation score (BMI SDS) instead of BMI to compare the risk of HBP in different groups with multivariate logistic models. Age- and sex-specific BMI SDS was calculated using the World Health Organization (WHO) child growth charts [25–27].

All $P$ values were two-sided and $P$ value of $<0.05$ were considered as statistically significant. SPSS for Windows (version 20.0, SPSS Inc, Chicago, IL, USA) was used for statistical analysis.

## Results

### Characteristic of the study population

A total of 28,288 children aged 6–17 years were included for the present study, 51.4% were boys, with median age of 10 years (IQR: 8–13 years), and the mean follow up period was 6.88 ±1.20 months. At the baseline, 12.2% of the children were categorized as GOB (15.3% and 9.0% for boys and girls, respectively), 15.5% had AOB (19.0% and 11.8% for boys and girls, respectively), and 9.7% had HBP (10.5% and 8.7% for boys and girls, respectively) (Table 1). The prevalence of obesity based on the international definition of child obesity developed by IOTF was 6.1% at the baseline (8.4% vs 3.6% for boys and girls, respectively) and 5.1% at the follow-up survey (7% vs 3% for boys and girls, respectively, S2 Table). The prevalence of

**Table 1. General characteristics of the study population.**

| Variables | Group | Boys (n = 14542) | | Girls (n = 13746) | | Total (N = 28288) | | P value |
|---|---|---|---|---|---|---|---|---|
| | | N/Median | %/IQR | N/Median | %/IQR | N/Median | %/IQR | |
| *Characteristics at the Baseline survey* | | | | | | | | |
| Area | urban | 9520 | 65.50% | 9035 | 65.70% | 18555 | 65.60% | 0.642 |
| | rural | 5022 | 34.50% | 4711 | 34.30% | 9733 | 34.40% | |
| Age(years) | | 10 | (8,13) | 10 | (8,14) | 10 | (8,13) | <0.001 |
| Height(cm) | | 142.85 | (131.05,163.15) | 145.55 | (130.6,157.5) | 143.9 | (131,159.2) | <0.001 |
| Weight(kg) | | 38.2 | (28.1,53) | 38 | (26.85,49) | 38.05 | (27.5,50.85) | <0.001 |
| BMI(kg/m$^2$) [a] | | 18 | (15.95,20.8) | 17.67 | (15.54,20.19) | 17.85 | (15.75,20.48) | <0.001 |
| Waist circumference(cm) | | 64.15 | (56.8,72.05) | 62.85 | (55.6,70) | 63.45 | (56.1,71) | <0.001 |
| WHtR[b] | | 0.44 | (0.41,0.48) | 0.43 | (0.41,0.47) | 0.43 | (0.41,0.47) | <0.001 |
| SBP(mmHg) | | 105 | (97,114) | 101 | (94,110) | 103 | (96,111) | <0.001 |
| DBP(mmHg) | | 67 | (60,71) | 65 | (60,70) | 66 | (60,71) | <0.001 |
| Fruits(servings/day)[c] | | 1 | (0.57,1.71) | 1 | (0.57,1.71) | 1 | (0.57,1.71) | <0.001 |
| Vegetables(servings/day) [c] | | 1.43 | (1,2) | 1.43 | (1,2) | 1.43 | (1,2) | <0.001 |
| SSB [d] (cup/day)[e] | | 0.2 | (0,0.57) | 0.14 | (0,0.29) | 0.14 | (0,0.43) | <0.001 |
| Vigorous physical activity(hour/day) | | 0.29 | (0.07,0.67) | 0.19 | (0.02,0.48) | 0.21 | (0.05,0.57) | <0.001 |
| Moderate physical activity(hour/day) | | 0.25 | (0,0.67) | 0.21 | (0,0.5) | 0.21 | (0,0.62) | <0.001 |
| General obesity | no | 12319 | 84.70% | 12515 | 91.00% | 24834 | 87.80% | <0.001 |
| | yes | 2223 | 15.30% | 1231 | 9.00% | 3454 | 12.20% | |
| Abdominal obesity | no | 11723 | 81.00% | 12071 | 88.20% | 23794 | 84.50% | <0.001 |
| | yes | 2754 | 19.00% | 1612 | 11.80% | 4366 | 15.50% | |
| HBP | no | 13012 | 89.50% | 12546 | 91.30% | 25558 | 90.30% | <0.001 |
| | yes | 1530 | 10.50% | 1200 | 8.70% | 2730 | 9.70% | |
| *Characteristics at the follow-up survey* | | | | | | | | |
| Age(years) | | 11 | (9,14) | 11 | (9,14) | 11 | (9,14) | <0.001 |
| Height(cm) | | 146.4 | (134.4,166) | 149 | (134,158.5) | 147.55 | (134.2,161) | <0.001 |
| Weight(kg) | | 40.6 | (29.5,55) | 40.35 | (28.5,50.6) | 40.5 | (29,52.55) | <0.001 |
| BMI(kg/m$^2$) | | 18.07 | (15.96,20.88) | 17.9 | (15.58,20.45) | 17.99 | (15.78,20.67) | <0.001 |
| Waist circumference(cm) | | 64 | (56.45,72.1) | 62 | (55,69) | 63 | (56,70.5) | <0.001 |
| WHtR | | 0.42 | (0.4,0.47) | 0.42 | (0.4,0.45) | 0.42 | (0.4,0.46) | <0.001 |
| SBP(mmHg)[f] | | 105 | (98,112) | 102 | (96,110) | 103 | (98,111) | <0.001 |
| DBP(mmHg)[g] | | 65 | (60,70) | 64 | (60,70) | 64 | (60,70) | <0.001 |
| General obesity | no | 12714 | 87.40% | 12668 | 92.20% | 25382 | 89.70% | <0.001 |
| | yes | 1828 | 12.60% | 1078 | 7.80% | 2906 | 10.30% | |
| Abdominal obesity | no | 12141 | 84.00% | 12523 | 91.40% | 24664 | 87.60% | <0.001 |
| | yes | 2320 | 16.00% | 1184 | 8.60% | 3504 | 12.40% | |
| HBP[h] | no | 13617 | 93.60% | 13019 | 94.70% | 26636 | 94.20% | <0.001 |
| | yes | 925 | 6.40% | 727 | 5.30% | 1652 | 5.80% | |
| *Change of obesity status from baseline to follow-up* | | | | | | | | |
| General obesity status change[i] | NN | 12185 | 83.80% | 12408 | 90.30% | 24593 | 86.90% | <0.001 |
| | YN | 529 | 3.60% | 260 | 1.90% | 789 | 2.80% | |
| | NY | 134 | 0.90% | 107 | 0.80% | 241 | 0.90% | |
| | YY | 1694 | 11.60% | 971 | 7.10% | 2665 | 9.40% | |

(*Continued*)

**Table 1.** (Continued)

| Variables | Group | Boys (n = 14542) | | Girls (n = 13746) | | Total (N = 28288) | | P value |
|---|---|---|---|---|---|---|---|---|
| | | N/Median | %/IQR | N/Median | %/IQR | N/Median | %/IQR | |
| Abdominal obesity status change[i] | NN | 11391 | 79.10% | 11798 | 86.40% | 23189 | 82.60% | <0.001 |
| | YN | 707 | 4.90% | 674 | 4.90% | 1381 | 4.90% | |
| | NY | 289 | 2.00% | 248 | 1.80% | 537 | 1.90% | |
| | YY | 2021 | 14.00% | 931 | 6.80% | 2952 | 10.50% | |

[a] BMI: body mass index.

[b] WHtR: waist-to-height ratio.

[c] A serving of fruit or vegetable is equivalent to 100 g.

[d] SBBs means sugar–sweetened beverage.

[e] A cup is equivalent to 250 ml.

[f] SBP: systolic blood pressure.

[g] DBP: diastolic blood pressure.

[h] HBP: high blood pressure.

[i] NN: non-obese at baseline and non-obese at follow-up;

NY: non-obese at baseline and obese at follow-up; YN: obese at baseline and non-obese at follow-up; YY: obese at baseline and obese at follow-up. IQR: inter quartile range.

obesity based on the WHO standard was 10.1% (14.8% vs 5.2% for boys and girls, respectively) at baseline and 8.8% at the follow-up survey (12.6% vs 4.7% for boys and girls, respectively, S2 Table).

At the follow-up survey, 10.3%, 12.4% and 5.8% of the participants had GOB, AOB and HBP, respectively. During the short-term follow-up period, for GOB, 86.9% of the children remained a non-GOB status, 9.4% of the children had persistent GOB, 2.8% converted from GOB to non-GOB, and 0.9% had newly developed GOB. Among the 3454 children with GOB at baseline, 77.2% of them had persistent GOB, and the conversion rate from GOB to non-GOB was 22.8% without intervention.

While for AOB, 82.6% of the children remained a non-AOB status, 10.5% of the children remained as AOB, 4.9% converted from AOB to non-AOB and 1.9% had newly developed AOB (Table 1).

### The BP levels and the risk of HBP in children with different obesity status change

Among children of different GOB status change groups, the baseline SBP and DBP were significantly different ($P<0.001$). SBP and DBP levels at follow-up, as well as the change of SBP/DBP during the short-term follow-up, of children in different groups were also significantly different ($P<0.05$). The risk of HBP was significantly different whenever baseline or follow-up in different GOB status change groups ($P<0.001$). Children with continuous GOB (YY group) had the highest prevalence of HBP (17.6%) at the follow-up survey, 12.9% for newly developed GOB group (NY group), 11.2% for children who had GOB before but not now (YN group), and 4.3% for the stable non-obese children (NN group). A similar pattern was also observed for AOB (Table 2).

The adjusted means of BP change with 95% confidence interval from baseline and follow-up were presented in Fig 1. The relationships between changes in BP and changes in BMI or WHtR were presented in the scatter plots (S2 Fig).

**Table 2. Blood pressure level and risk of high blood pressure of different pattern of general or abdominal obesity.**

| Variables | | NN | | YN | | NY | | YY | | P value |
|---|---|---|---|---|---|---|---|---|---|---|
| | | N/Mean | %/SD | N/Mean | %/SD | N/Mean | %/SD | N/Mean | %/SD | |
| *General obesity status change* | | | | | | | | | | |
| $SBP_{baseline}$ | | 103.36 | 11.54 | 106.44 | 13.01 | 108.46 | 11.97 | 111.29 | 12.77 | <0.001 |
| $SBP_{follow-up}$ | | 103.49 | 10.84 | 107.21 | 11.56 | 110.58 | 11.31 | 112.69 | 12.32 | <0.001 |
| SBP change | | 0.12 | 12.16 | 0.77 | 13.59 | 2.12 | 12.71 | 1.4 | 13.84 | <0.001 |
| $DBP_{baseline}$ | | 65.73 | 8.44 | 67.85 | 9.3 | 68.5 | 8.94 | 70.36 | 8.89 | <0.001 |
| $DBP_{follow-up}$ | | 65.1 | 8.05 | 67.35 | 8.46 | 69.61 | 7.87 | 70.59 | 8.63 | <0.001 |
| DBP change | | -0.63 | 10.07 | -0.5 | 10.96 | 1.11 | 10.75 | 0.23 | 10.89 | <0.001 |
| $HBP_{baseline}$ | no | 22617 | 92.00% | 653 | 82.80% | 205 | 85.10% | 2083 | 78.20% | <0.001 |
| | yes | 1976 | 8.00% | 136 | 17.20% | 36 | 14.90% | 582 | 21.80% | |
| $HBP_{follow-up}$ | no | 23528 | 95.70% | 701 | 88.80% | 210 | 87.10% | 2197 | 82.40% | <0.001 |
| | yes | 1065 | 4.30% | 88 | 11.20% | 31 | 12.90% | 468 | 17.60% | |
| *Abdominal obesity status change* | | | | | | | | | | |
| $SBP_{baseline}$ | | 103.21 | 11.52 | 105.42 | 11.97 | 105.13 | 11.88 | 111.47 | 12.46 | <0.001 |
| $SBP_{follow-up}$ | | 103.28 | 10.82 | 106.19 | 10.98 | 108.86 | 11.52 | 112.46 | 11.83 | <0.001 |
| SBP change | | 0.07 | 12.14 | 0.77 | 12.85 | 3.73 | 13.38 | 0.99 | 13.66 | <0.001 |
| $DBP_{baseline}$ | | 65.63 | 8.43 | 67.14 | 8.88 | 67.41 | 8.69 | 70.45 | 8.74 | <0.001 |
| $DBP_{follow-up}$ | | 64.96 | 8.02 | 67.12 | 8.17 | 68.33 | 8.61 | 70.41 | 8.48 | <0.001 |
| DBP change | | -0.67 | 10.05 | -0.02 | 10.8 | 0.92 | 10.53 | -0.05 | 10.8 | <0.001 |
| $HBP_{baseline}$ | no | 21343 | 92.00% | 1197 | 86.70% | 483 | 89.90% | 2325 | 78.80% | <0.001 |
| | yes | 1846 | 8.00% | 184 | 13.30% | 54 | 10.10% | 627 | 21.20% | |
| $HBP_{follow-up}$ | no | 22198 | 95.70% | 1253 | 90.70% | 478 | 89.00% | 2506 | 84.90% | <0.001 |
| | yes | 991 | 4.30% | 128 | 9.30% | 59 | 11.00% | 446 | 15.10% | |

SBP: systolic blood pressure. DBP: diastolic blood pressure. HBP: high blood pressure. NN: non-obese at baseline and non-obese at follow-up; NY: non-obese at baseline and obese at follow-up; YN: obese at baseline and non-obese at follow-up; YY: obese at baseline and obese at follow-up.

## Association between obesity status change and risk of HBP

Association between GOB or AOB status change groups and risk of HBP in four logistic regression models were presented in Table 3. In the crude models, no matter for GOB or AOB, the children with continuous obese status (YY group), or newly developed obesity group (NY group), or obese before but not now (YN group) had significantly higher risks of HBP compared with the sustained non-obese children (NN group, $P<0.001$). When further adjusted for different lifestyle covariates (model 3 or model 4), the children in YN group, YY group, or NY group still had significantly higher risks of HBP compared with the sustained non-obese children (NN group, $P<0.001$). In the full adjusted models (model 4), ORs (95%CI) for the YN group, NY and YY group of GOB were 2.72 (95%CI: 2.03~3.63), 3.35 (95%CI: 1.99~5.65) and 5.03 (95%CI: 4.32~5.86), respectively. Similar results were found for the AOB status change, the ORs(95%CI) for the YN group, NY and YY group were 2.24 (95%CI: 1.76~2.85), 3.42 (95%CI: 2.36~4.95), and 4.79 (95%CI: 4.09~5.60), respectively.

Sensitivity analyses generated similar results (S3 Table). Further adjustment of height change or baseline HBP status did not change our results. The estimates (ORs) are very similar as before additional adjustment and still statistically significant for the NY and YY groups.

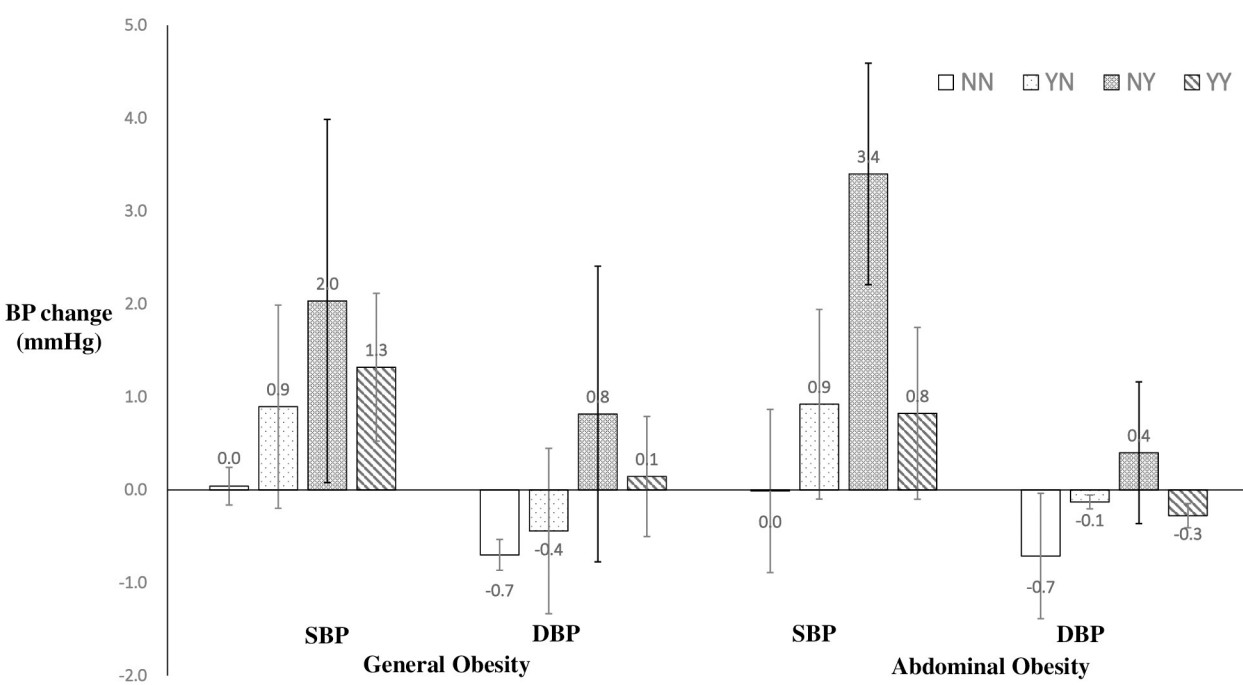

**Fig 1. Adjusted means and 95% confidence interval of blood pressure change from baseline to the follow-up in different general or abdominal obesity status change groups.** NN: non-obese at baseline and non-obese at follow-up; NY: non-obese at baseline and obese at follow-up; YN: obese at baseline and non-obese at follow-up; YY: obese at baseline and obese at follow-up. Means were adjusted for age, gender, area, BMI, fruit consumption, vegetable consumption, sugar-sweetened beverage intake and physical activities.

## Association between risk of HBP and different quintiles of BMI or WHtR change in the obese children

For children with GOB, when categorized as quintiles by the BMI change from baseline to follow-up, only quintile 1 (BMI change $\leq$-0.88 kg/m$^2$) and quintile 2 (BMI change -0.88~ -0.21 kg/m$^2$) were significantly associated with reduced risk of HBP (OR = 0.62, 95%CI: 0.46~0.84 vs OR = 0.68, 95%CI: 0.50~0.91 for quintile 1 and quintile 2, respectively). A reduction of at least 0.21~0.88 kg/m$^2$ of BMI (only 0.86%~3.59% of the baseline BMI, baseline BMI = 24.49 kg/m$^2$,

**Table 3. Association between risk of high blood pressure and different general or abdominal obesity status change.**

| Obesity type | Group[a] | Model 1 | | Model 2 | | Model 3 | | Model 4 | |
|---|---|---|---|---|---|---|---|---|---|
| | | OR(95%CI) | P | OR(95%CI) | P | OR(95%CI) | P | OR(95%CI) | P |
| General obesity | NN | 1(Ref.) | | 1(Ref.) | | 1(Ref.) | | 1(Ref.) | |
| | YN | 2.77(2.20~3.49) | <0.001 | 2.53(2.01~3.19) | <0.001 | 2.53(2.00~3.20) | <0.001 | 2.72(2.03~3.63) | <0.001 |
| | NY | 3.26(2.23~4.78) | <0.001 | 3.17(2.16~4.65) | <0.001 | 3.15(2.13~4.66) | <0.001 | 3.35(1.99~5.65) | <0.001 |
| | YY | 4.71(4.19~5.29) | <0.001 | 4.47(3.97~5.04) | <0.001 | 4.79(4.23~5.43) | <0.001 | 5.03(4.32~5.86) | <0.001 |
| Abdominal obesity | NN | 1(Ref.) | | 1(Ref.) | | 1(Ref.) | | 1(Ref.) | |
| | YN | 2.29(1.89~2.77) | <0.001 | 2.18(1.80~2.65) | <0.001 | 2.03(1.67~2.48) | <0.001 | 2.24(1.76~2.85) | <0.001 |
| | NY | 2.77(2.09~3.65) | <0.001 | 2.77(2.10~3.66) | <0.001 | 3.06(2.30~4.07) | <0.001 | 3.42(2.36~4.95) | <0.001 |
| | YY | 3.99(3.54~4.49) | <0.001 | 3.93(3.48~4.43) | <0.001 | 4.48(3.94~5.09) | <0.001 | 4.79(4.09~5.60) | <0.001 |

[a]: NN: non-obese at baseline and non-obese at follow-up;

NY: non-obese at baseline and obese at follow-up; YN: obese at baseline and non-obese at follow-up; YY: obese at baseline and obese at follow-up. Model 1 is the crude model. Model 2 is adjusted for age and gender. Model 3 is further adjusted for province, and area. Model 4 is further adjusted for fruits consumption, vegetable consumption, sugar-sweetened beverage intake, vigorous and moderate physical activity.

**Table 4. Association between risk of high blood pressure and different quintile of BMI or WHtR change in the obese children.**

| Variables | Group | Model 1 | | Model 2 | |
|---|---|---|---|---|---|
| | | OR(95%CI) | *P* | OR(95%CI) | *P* |
| BMI change (kg/m$^2$) | Quintile 1(≤-0.88) | 0.70(0.52~0.93) | 0.014 | 0.62(0.46~0.84) | 0.002 |
| | Quintile 2(-0.88~-0.21) | 0.71(0.53~0.95) | 0.020 | 0.68(0.50~0.91) | 0.010 |
| | Quintile 3(-0.21~0.31) | 0.77(0.58~1.02) | 0.073 | 0.77(0.58~1.03) | 0.081 |
| | Quintile 4(0.31~0.93) | 1.02(0.78~1.34) | 0.891 | 1.03(0.78~1.36) | 0.828 |
| | Quintile 5(>0.93) | 1(Ref.) | | 1(Ref.) | |
| WHtR change | Quintile 1(≤-0.045) | 0.64(0.49~0.85) | 0.002 | 0.48(0.35~0.66) | <0.001 |
| | Quintile 2(-0.045~ -0.024) | 0.70(0.53~0.93) | 0.012 | 0.69(0.52~0.93) | 0.013 |
| | Quintile 3(-0.024~ -0.009) | 0.69(0.52~0.91) | 0.009 | 0.71(0.53~0.94) | 0.016 |
| | Quintile 4(-0.009~ 0.007) | 1.02(0.79~1.32) | 0.888 | 1.03(0.79~1.33) | 0.853 |
| | Quintile 5(>0.007) | 1(Ref.) | | 1(Ref.) | |

Model 1 is the crude model. Model 2 is adjusted for age, gender, province, and area. BMI: body mass index. WHtR: waist-to-height ratio.

0.21/24.49 = 0.86%, and 0.88/24.49 = 3.59%) in children with GOB (baseline BMI = 24.49 kg/m$^2$) could be significantly beneficial to the BP profile compared with the reference group (Table 4).

For children with AOB, when categorized as quintiles by the WHtR change from baseline to follow-up, children in quintile 1 (WHtR change ≤-0.045), quintile 2 (WHtR change -0.045~ -0.024) and quintile 3(WHtR change -0.024~ -0.009) were significantly associated with reduced risks of HBP (OR = 0.48, 95%CI: 0.35~0.66, OR = 0.69, 95%CI: 0.52~0.93 and OR = 0.71, 95%CI: 0.53~0.94 for quintile 1, quintile 2 and quintile 3, respectively). A reduction of at least 0.009~0.024 (1.66%~4.42% of the baseline WHtR, baseline WHtR = 0.5429, 0.009/0.5429 = 1.66%, 0.024/0.5429 = 4.42%) of WHtR in children with AOB (baseline WHtR = 0.5429) could be significantly beneficial to the BP profile (Table 4).

We also categorized the BMI or WHtR change as ten equal groups with percentile of $P_{10}$, $P_{20}$, $P_{30}$, $P_{40}$, $P_{50}$, $P_{60}$, $P_{70}$, $P_{80}$ and $P_{90}$ as cutoff points. Similar results were observed (S4 Table). We also used two international definitions of child obesity (ITOF standard and WHO standard) to test the associations between BP level or HBP risk and obesity change pattern, similar results were observed (S5–S7 Tables). In addition, BMI SDS was used instead of BMI to test the association of different quintiles of BMI SDS change and risk of HBP, similar results were observed (S8 Table).

## Discussion

In this short-term follow-up study with 28,288 children aged 6–17 years, we demonstrated the obesity status change during the short-term follow up, and compared the risk of HBP in different obesity status change groups. We found that children with persistent obesity, newly developed obesity or those converting from obese to non-obese had significantly higher risk of HBP compared with persistent non-obese children no matter for GOB or AOB. At the same time, we found that a reduction of 0.86%-3.59% of the baseline BMI in children with GOB, or reduction of 1.66%-4.42% of WHtR in children with AOB could be significantly beneficial to the BP profile.

### GOB or AOB prevalence in children

During recent years, pediatric obesity has become an epidemic in China. Previous survey reported that the prevalence of overweight and obesity of 7-18-year-old children was 19.4% in

2014, while the corresponding prevalence was only 1.24% in 1985 [28]. According to our study, without intervention, about 9.4% children have persistent GOB, which is more than double of the rate (3.9%) in Ziyab study among UK children [29], obviously higher than Berentzen' study in Dutch [30]. Berentzen's study reported that the rate of persistent over-weight in children is 4%. The difference between our findings with these two European studies may due to the genetic background, or the different lifestyles, or various obesogenic socio-economic environment. But obviously, this serious situation of children requires united policies, efforts from the government, school and family to tackle it.

Notably, our study identified 10.5% of the children who had persistent AOB without intervention. As far as we are concerned, no previous study reported the rate of persistent AOB with a longitudinal design among Chinese children. Undoubtedly, WHtR as an important indicator for abdominal fat accumulation, 10.5% is a high prevalence which needs united efforts to tackle this public health problem, including inter-sectional actions from government, policies in schools, and efforts from the whole families [31, 32]. Since many epidemiology studies demonstrated that AOB is closely related with risk of CVDs and diabetic related complications [33–35]. Given the big population in China, without proper interventions or control policies for pediatric obesity, it will pose a great cardiovascular disease burden related to obesity in the near future.

## Different GOB or AOB status change with risk of HBP

Along with the epidemic of obesity the prevalence of HBP in children has also increased in recent years, cross-sectional association between obesity and HBP has been well documented, but whether the obesity status change without intervention would have an immediate impact on HBP among children has not been concluded [36–39]. Our study found that compared with the persistent non-obese children, those once obese children have a doubled risk of HBP, while the newly developed obesity children tripled the HBP risk of the reference group. Our finding is similar to a recent systematic review about weight change from childhood to adulthood and later life cardiovascular risk factors [37], showing that the related risks reduced when children with obesity turned to be non-obese, and even eliminated when obesity turned to normal weight. But at the same time there are inconsistent studies, for example, Michael's study showed that every single period of obesity could be detrimental to the BP profile no matter the weight status in other periods [36]. This discrepancy may be due to the racial or ethnicity differences [39]. Surprisingly, in our study the newly developed obesity will sharply increase (more than triple) the risk of HBP almost the same magnitude of that in persistent obese children. This is consistent with a prospective study conducted by Li et al., their study found that excessive BMI gain, especially recently BMI increase, would significantly increase BP [40].

Regarding the association between AOB status change and risk of HBP, we used WHtR cut-off point (0.5) defined AOB, which is a simple but predictive indicator often used to measure abdominal fat accumulation [13, 41, 42]. Previous longitudinal and cross-sectional studies investigated the associations between WHtR and elevated BP [35, 42–44], and found positive relation between WHtR or AOB and BP or elevated BP, but no studies investigated the longitudinal association between the change pattern of AOB status and risk of HBP. Until now, information about AOB status change in childhood is really scarce. We found the persistent or newly developed AOB can cause the same level of risk of HBP. Therefore, it is also crucial to monitor the abdominal fat accumulation, then prevent and control the development or persistence of AOB.

In addition, the sensitivity analyses generated similar results. Further adjustment of height change or baseline HBP status did not change our results. The estimates (ORs) are very similar as before additional adjustment and still statistically significant for the NY and YY groups, which shows that our findings are relatively robust and stable.

### Association between risk of HBP and different quintiles of BMI or WHtR change in the obese children

Study of childhood obesity intervention, 5% of the BMI reduction is significantly associated with more favorable cardiometabolic profiles (lipid, insulin sensitivity and BP) [8]. A reduction of only 0.86%-3.59% of BMI could lead to beneficial effect on HBP in our study. However, a recent meta-analysis of 54 lifestyle intervention studies in pediatric populations showed that reduction of >1 mean BMI SDS is related to improved SBP [7]. Considering the difference of race, and age in different studies, it is hard to compare or make a final quantification of the minimum favorable BMI reduction for metabolic health. But given the short-term reduction of BMI (0.86%-3.59%) or WHtR (1.66%-4.42%) among obese children could immediately have a significant effect of reducing the risk of HBP, this is encouraging for weight management in children.

### Strengths and limitations

This is a large-scale study with random sampling method covering 7 provinces in China, our sample is representative and the findings could be generated to children of China. Also, we longitudinally measured the anthropometric indices and BP for all the participants, the associations between obesity status change and risk of HBP are stronger and more evident than the cross-sectional associations. Also, a variety of sensitivity analyses generated similar results, this indicates our findings are robust. For example, additional adjustment of height change (reflecting the growth effect) did not substantially change our results, suggesting that the normal growth is not likely to have considerable effect on our results. In addition, the associations between different obesity status change and risk of HBP are verified with both GOB and AOB.

There are limitations in our study. First, our study is only a short-term follow up focused in childhood, but it is also significant because even in a short-term follow-up the newly developed obesity could be unfavorable for BP profile as persistent obesity. This means if the obesity status continues, the detrimental effects on HBP could track into later life, implying that early life intervention for the obesity is quite important for these children. Second, the BP measurement is only one-occasion measurement due to the limited founding of the study, future studies involving three-occasion BP measurement could be done to verify the findings.

### Conclusion

To conclude, we examined GOB and AOB status change among Chinese children in a short-term follow-up, and the impact of the adiposity status change on BP profiles. Compared with persistent non-GOB children, children with persistent GOB had the highest risk of HBP (more than 5-fold), children with newly developed GOB had second higher HBP risk (more than triple), those converting from GOB to non-GOB has a more than double HBP risk compared with the reference group. Similar findings were also found for the relation between AOB status change with HBP risk. Reduction of 0.86%-3.59% of BMI or 1.66%-4.42% of WHtR among GOB or AOB children, respectively, during about half a year could be significantly beneficial to the BP profile. Our results highlight the importance of preventing and controlling GOB and AOB in childhood, to reduce the cumulative detrimental effects of high BP since early life.

## Supporting information

**S1 Table. Body mass index reference norm for screening overweight and obesity among Chinese children aged 6–18 years (kg/m$^2$).**
(DOCX)

**S2 Table. The prevalence of obesity in children based on two international definitions of child obesity*.**
(DOCX)

**S3 Table. Association between risk of HBP and different obesity status change with adjustment of other potential covariates.**
(DOCX)

**S4 Table. Association between risk of high blood pressure and different groups (ten equal groups) of BMI or WHtR change in the obese children.**
(DOCX)

**S5 Table. Blood pressure level and risk of high blood pressure of different pattern of general obesity based on two international definitions of child obesity.**
(DOCX)

**S6 Table. Association between risk of high blood pressure and different general or abdominal obesity status change based on two international definitions of child obesity.**
(DOCX)

**S7 Table. Association between risk of high blood pressure and different quintile of BMI change in the obese children based on two international definitions of child obesity.**
(DOCX)

**S8 Table. Association between risk of high blood pressure and different quintile of BMI SDS change in the obese children.**
(DOCX)

**S1 Fig. The size of one serving of fruit or vegetable (as the size of a fist of ordinary adult).**
(DOCX)

**S2 Fig. The relationships between changes in blood pressures and changes in body mass index (BMI) or waist-to-height ratio (WHtR).**
(DOCX)

**S1 File. Questionnaire about dietary factors and physical activities in children and adolescents (including S1 Fig).**
(DOC)

## Acknowledgments

We would like to thank all the children for their participation in our study, also we would like to thank all the head teachers of the schools for their help and cooperation in the research.

## Author Contributions

**Conceptualization:** Yi-de Yang, Ming Xie, Zhiyong Zou, Xiangli Ye, Xiuqin Hong, Qiu Xiao.

**Data curation:** Yi-de Yang, Bin Dong, Zhenghe Wang.

**Formal analysis:** Yi-de Yang, Yuan Zeng, Shuqian Yuan, Haokai Tang, Qiu Xiao.

**Funding acquisition:** Yi-de Yang, Haokai Tang, Xiangli Ye.

**Investigation:** Yi-de Yang, Zhenghe Wang.

**Methodology:** Yi-de Yang.

**Project administration:** Bin Dong, Jun Ma.

**Resources:** Jun Ma.

**Software:** Yi-de Yang, Bin Dong, Xiuqin Hong, Jun Ma.

**Supervision:** Jun Ma.

**Validation:** Yi-de Yang.

**Writing – original draft:** Yi-de Yang, Ming Xie, Yanhui Dong.

**Writing – review & editing:** Yi-de Yang, Ming Xie, Yuan Zeng, Shuqian Yuan, Haokai Tang, Yanhui Dong, Zhiyong Zou, Bin Dong, Zhenghe Wang, Xiangli Ye, Xiuqin Hong, Qiu Xiao, Jun Ma.

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
