## [Decision Letter · Decision Letter 0]

21 Jul 2021

PONE-D-21-04822

Impact of short-term change of adiposity on risk of high blood pressure in children: Results from a follow-up study in China.

PLOS ONE

Dear Dr. Yang,

Thank you for submitting your manuscript to PLOS ONE. After careful consideration, we feel that it has merit but does not fully meet PLOS ONE’s publication criteria as it currently stands. Therefore, we invite you to submit a revised version of the manuscript that addresses the points raised during the review process.

We look forward to receiving your revised manuscript.

Kind regards,

Raffaella Buzzetti, M.D.

Academic Editor

PLOS ONE

Journal Requirements:

2. Please review your reference list to ensure that it is complete and correct. If you have cited papers that have been retracted, please include the rationale for doing so in the manuscript text, or remove these references and replace them with relevant current references. Any changes to the reference list should be mentioned in the rebuttal letter that accompanies your revised manuscript. If you need to cite a retracted article, indicate the article’s retracted status in the References list and also include a citation and full reference for the retraction no

3. Please ensure you have included the registration number for the clinical trial referenced in the manuscript. Moreover, please clarify 1) the inclusion criteria used for recruitment, as we note that the trial registry indicates that children aged 7-18, not 6-17, were included; 2) where the study took place; 3) the length of the follow-up (which in the trial protocol is set at 9 months).

4. We suggest you thoroughly copyedit your manuscript for language usage, spelling, and grammar. If you do not know anyone who can help you do this, you may wish to consider employing a professional scientific editing service. 

A clean copy of the edited manuscript (uploaded as the new *manuscript* file).

We would like to thank all the children for their participation in our study, also we would like to thank all the head teachers of the schools for their help and cooperation in the research. This research was funded by the National Natural Science Foundation of China (81903336, YY), the Hunan Provincial Natural Science Foundation of China (2019JJ50376, YY), Scientific Research Project of Hunan Health Committee (202112031516,YY), and a Project Supported by Scientific Research Fund of Hunan Provincial Education Department (18A0028, XY). The funders had no role in the design, analysis or writing of this article.

This research was funded by the National Natural Science Foundation of China (81903336, YY), the Hunan Provincial Natural Science Foundation of China (2019JJ50376, YY), Scientific Research Project of Hunan Health Committee (202112031516,YY), and a Project Supported by Scientific Research Fund of Hunan Provincial Education Department (18A0028, XY). The funders had no role in the design, analysis or writing of this article.

7. We note that you have indicated that data from this study are available upon request. PLOS only allows data to be available upon request if there are legal or ethical restrictions on sharing data publicly. For more information on unacceptable data access restrictions, please see http://journals.plos.org/plosone/s/data-availability#loc-unacceptable-data-access-restrictions. 

8. Your ethics statement should only appear in the Methods section of your manuscript. If your ethics statement is written in any section besides the Methods, please delete it from any other section. 

9. Please upload a copy of Supporting Information Table S3 which you refer to in your text on page 16. 

Reviewers' comments:

Reviewer's Responses to Questions

**Comments to the Author**

1. Is the manuscript technically sound, and do the data support the conclusions?

Reviewer #1: Yes

Reviewer #2: Yes

Reviewer #3: Yes

2. Has the statistical analysis been performed appropriately and rigorously? 

Reviewer #1: Yes

Reviewer #2: Yes

Reviewer #3: Yes

3. Have the authors made all data underlying the findings in their manuscript fully available?

Reviewer #1: Yes

Reviewer #2: Yes

Reviewer #3: Yes

4. Is the manuscript presented in an intelligible fashion and written in standard English?

Reviewer #1: Yes

Reviewer #2: Yes

Reviewer #3: Yes

5. Review Comments to the Author

Reviewer #1: The statistics used in this paper, both univariate and multivariate, are fairly routine and the sample size is certainly large enough. The paper, given the size of the sample, is more descriptive than inferential and the conclusions follow from the analyses performed.

Reviewer #2: An important issue, the one the Authors decided to confront with, and given the big sample they could put together, I think their results are worth being widespread.

Here just a few remarks:

- It's a pity you used "standard “BMI Reference for Screening for overweight and obesity among school-age children and adolescents” published by National Health Commission of the People‘s Republic of China" instead of more internationally accepted ones: thinking of future research, it makes difficult to compare your results with other obtained in different countries. I believe that this point deserves to be added to the limitations of the study, and possibly counter-explained.

- At lines 224-225 the sentence "A reduction of at least 0.21~0.88 kg/m2 of BMI (only 0.86%~3.59% of the baseline BMI) in children with GOB..." is not clear to me; specifically, what do you mean with "(only 0.86%~3.59% of the baseline BMI)"?

- At lines 274-276, the meaning of the sentence "those converted from obese to non-obese doubles the HBP risk compared with the persistent non-obese children" is clear enough, but I believe it should be re-phrased.

- There is some other minor revision of English to be made; in the attached file I highlighted the line numbers where some correction is needed, IMHO. There are possibly more than those ones, but they are the ones I easily spotted.

Reviewer #3: 1. Did the Authors use some exclusion criteria to enrol the subjects?

2. All the investigated variables were normally distributed? Which kind of test the Authors used to verify data normality?

3. It would be better use BMI-SDS instead of BMI considering the age of subjects.

4. Why the Authors decided to categorize the adiposity parameter (BMI or WHtR) change during the follow-up as quintiles?

5. In logistic regression, in addition to OR they have to show and mention also the goodness of fit test of the model. To adjust a regression for multiple variables without to verify if these variable are possible confounders is not correct. Mantel–Haenszel method could be used to verify if a variable is a possible confounder.

6. Could the Authors clarify the sentence a” reduction of at least 0.21-~0.88 kg of BMI (only 0.86%~3.59% of the baseline BMI) in children with GOB (baseline BMI=24.49 kg) at lines 224. It is not very clear: only 0.86%~3.59% of the baseline BMI.

6. PLOS authors have the option to publish the peer review history of their article (what does this mean?). If published, this will include your full peer review and any attached files.

Reviewer #1: No

Reviewer #2: **Yes: **Andrea Vania

Reviewer #3: No

---

## [Author Response · Author response to Decision Letter 0]

17 Aug 2021

Response letter to the Editors’ suggestions

Response: Thanks for the suggestion. We have revised our paper according to PLOS ONE’s style requirements.

2. Please review your reference list to ensure that it is complete and correct. If you have cited papers that have been retracted, please include the rationale for doing so in the manuscript text, or remove these references and replace them with relevant current references. Any changes to the reference list should be mentioned in the rebuttal letter that accompanies your revised manuscript. If you need to cite a retracted article, indicate the article’s retracted status in the References list and also include a citation and full reference for the retraction no

Response: Thanks for the suggestion. We have checked all the references in the manuscript, no cited papers have been retracted.

3. Please ensure you have included the registration number for the clinical trial referenced in the manuscript. Moreover, please clarify 1) the inclusion criteria used for recruitment, as we note that the trial registry indicates that children aged 7-18, not 6-17, were included; 2) where the study took place; 3) the length of the follow-up (which in the trial protocol is set at 9 months).

Response: Thanks for the suggestion. We have added the registration number for this national multi-centered cluster randomized controlled trial (https://www.clinicaltrials.gov/, Registration number: NCT02343588) in the Method Section (Line 89).

1) the inclusion criteria used for recruitment

For the inclusion criteria used for our recruitment, the inclusion and exclusion criteria were listed as follows:

Inclusion Criteria:

Grade of 1 to 5 in primary schools and grade of 6 to 8 and 10 to 11 in secondary schools.

Exclusion Criteria:

Suffering or any history of disabilities, cardiovascular diseases, asthma, metabolic diseases, etc.

Actually, before the study began, we planned to recruit grade 1 to 5 in primary schools, grade of 6 to 8 and 10 to 11 in secondary schools. For students in grade 1 to 5 in primary schools, the age of them were supposed to be 7-11 years old. But actually, in the real-life study, some children aged 6 years old also started primary school and the intervention is school based. Therefore, there is slight difference in the age range in the registration trial (7-18 years old) and the present study (6-18 years old), also the age range of our paper is same with another paper about the effectiveness of this lifestyle intervention study (1).

[1] Dong Y, Zou Z, Wang H, Dong B, Hu P, Ma Y et al. National School-Based Health Lifestyles Intervention in Chinese Children and Adolescents on Obesity and Hypertension. Front Pediatr. 2021; 9(615283. https://doi.org/10.3389/fped.2021.615283.

2) where the study took place

The study was conducted in 7 provinces (including Hunan, Liaoning, Ningxia, Guangdong, Shanghai, Chongqing, and Tianjin) in China. We have added the description of study location in the Method Section (Line 98).

3) the length of the follow-up (which in the trial protocol is set at 9 months)

The length of follow up was 6.88 months (mean value), which is shorter than the length of follow up described in the study registry (9 months). This is because our study was conducted in 7 provinces (a total of 94 schools), the length of baseline physical examination and survey was almost a month, and the follow-up physical examination and survey also take a month to finish, so the range of the follow-up is set at 9 months, but for every participant at different schools the length of follow-up might be slightly different, and the mean value of follow-up is about 7 months.

We have added more description of the length of follow-up and age of the study population in the Method section as follows (Line 104-119).

For the enrollment of subjects, the inclusion criteria were as follows: grade of 1 to 5 in primary schools and grade of 6 to 8 and 10 to 11 in secondary schools. And the exclusion criteria were as follows: Suffering or any history of disabilities, cardiovascular diseases, asthma, etc. For students in grade 1 to 5 in primary schools, the age of them were supposed to be 7-11 years old. But actually, in the real life, some children aged 6 years old also started primary school. Therefore, the age range in the registration trial (7-18 years old) is slightly different that of the present study (6-18 years old), also the age range of our paper is same with another paper about the effectiveness of this lifestyle intervention study. The length of follow up was 6.88 months (mean value), which is shorter than the length of follow up described in the study registry (9 months). This is because our study was conducted in 7 provinces (a total of 94 schools), the length of baseline physical examination and survey was almost a month, and the follow-up physical examination and survey also take a month to complete, so the range of the follow-up is set at 9 months, but for every participant at different schools the length of follow-up might be slightly different, and the mean value of follow-up is 6.88 months.

4. We suggest you thoroughly copyedit your manuscript for language usage, spelling, and grammar. If you do not know anyone who can help you do this, you may wish to consider employing a professional scientific editing service.

Response: Thanks for the suggestion. We have thoroughly copyedited our manuscript for language usage, spelling, and grammar.

Response: Thanks for the suggestion. We have revised the “Funding information” and “Financial Disclosure” accordingly.

We would like to thank all the children for their participation in our study, also we would like to thank all the head teachers of the schools for their help and cooperation in the research. This research was funded by the National Natural Science Foundation of China (81903336, YY), the Hunan Provincial Natural Science Foundation of China (2019JJ50376, YY), Scientific Research Project of Hunan Health Committee (202112031516,YY), and a Project Supported by Scientific Research Fund of Hunan Provincial Education Department (18A0028, XY). The funders had no role in the design, analysis or writing of this article.

This research was funded by the National Natural Science Foundation of China (81903336, YY), the Hunan Provincial Natural Science Foundation of China (2019JJ50376, YY), Scientific Research Project of Hunan Health Committee (202112031516,YY), and a Project Supported by Scientific Research Fund of Hunan Provincial Education Department (18A0028, XY). The funders had no role in the design, analysis or writing of this article.

Response: Thanks for the suggestion. We have deleted the funding information in the Acknowledgments Section. Please help revise the funding information in the online submission as follows:

Funding

This research was funded by the National Natural Science Foundation of China (81903336), the Hunan Provincial Natural Science Foundation of China (2019JJ50376 & 2020JJ5386), Scientific Research Project of Hunan Provincial Health Commission (202112031516), Key Project of Hunan Provincial Science and Technology Innovation (2020SK1015-3) and a Project Supported by Scientific Research Fund of Hunan Provincial Education Department (18A0028). The funders had no role in the study design, data collection, data analysis, writing of this article or interpreting the results.

7. We note that you have indicated that data from this study are available upon request. PLOS only allows data to be available upon request if there are legal or ethical restrictions on sharing data publicly. For more information on unacceptable data access restrictions, please see http://journals.plos.org/plosone/s/data-availability#loc-unacceptable-data-access-restrictions. 

Response: Thanks for the suggestion. The datasets generated and/or analyzed during the present study

are not publicly available, since ethics approval and participants’ consent does not allow public sharing

of data, only available from the institute (Email: bindong@bjmu.edu.cn) upon reasonable request.

8. Your ethics statement should only appear in the Methods section of your manuscript. If your ethics statement is written in any section besides the Methods, please delete it from any other section. 

Response: Thanks for the suggestion. We have revised it accordingly.

9. Please upload a copy of Supporting Information Table S3 which you refer to in your text on page 16. 

Response: Thanks for the suggestion. We have added S3 Table in the Supplementary file 1.

Response letter for Reviewers

Reviewer #1: The statistics used in this paper, both univariate and multivariate, are fairly routine and the sample size is certainly large enough. The paper, given the size of the sample, is more descriptive than inferential and the conclusions follow from the analyses performed.

Response: Thanks for the comment. We have revised our paper according to your comment throughout the manuscript.

Reviewer #2: An important issue, the one the Authors decided to confront with, and given the big sample they could put together, I think their results are worth being widespread.

Here just a few remarks:

- It's a pity you used "standard “BMI Reference for Screening for overweight and obesity among school-age children and adolescents” published by National Health Commission of the People‘s Republic of China" instead of more internationally accepted ones: thinking of future research, it makes difficult to compare your results with other obtained in different countries. I believe that this point deserves to be added to the limitations of the study, and possibly counter-explained.

Response: We thank the reviewer for this comment. According to the suggestion, we have added more analysis based on two more internationally accepted standard definition for child overweight and obesity (developed by Cole T.J), which involving six large nationally representative cross sectional growth studies (Brazil, Great Britain, Hong Kong, the Netherlands, Singapore, and the United States) [1], which was also used in previous studies [2-4]. And also similar results based on the World Health Organization (WHO) standard were observed. Although lower prevalence of obesity was observed when two international definitions for child obesity was used (S2 Table), the associations between different obesity patterns with BP level or HBP risk were observed were similar with the results when Chinese BMI Reference was used (S5 Table, S6 Table, S7 Table). We have added these results as part of the sensitivity analysis, and the description of the method and results were added the text (Line 195-198, 212-217 and Line 291-293) as follows. 

In the Method section (Line 195-198)

(4) We used an international definition of child obesity developed by the International Obesity Task Force (IOTF) and the World Health Organization (WHO) standard of obesity for 5-19 years old children to test the associations[18-24].

In the result section (Line 212-217 and line 291-293)

The prevalence of obesity based on the international definition of child obesity developed by IOTF was 6.1% at the baseline (8.4% vs 3.6% for boys and girls, respectively) and 5.1% at the follow-up survey (7% vs 3% for boys and girls, respectively, S2 Table). The prevalence of obesity based the WHO standard was 10.1% (14.8% vs 5.2% for boys and girls, respectively) at baseline and 8.8% at the follow-up survey (12.6% vs 4.6% for boys and girls, respectively, S2 Table).

We also used two international definitions of child obesity (ITOF standard and WHO standard) to test the associations between BP level or HBP risk and obesity change pattern, similar results were observed (S5 Table, S6 Table, and S7 Table). 

18. Cole TJ, Bellizzi MC, Flegal KM, Dietz WH. Establishing a standard definition for child overweight and obesity worldwide: international survey. BMJ. 2000; 320: 1240-1243. 

19. Deren K, Wyszynska J, Nyankovskyy S, Nyankovska O, Yatsula M, Luszczki E, Sobolewski M, Mazur A: Assessment of body mass index in a pediatric population aged 7-17 from Ukraine according to various international criteria-A cross-sectional study. PLoS One 2020, 15(12):e0244300.

20. Oosterhoff M, Over EAB, van Giessen A, Hoogenveen RT, Bosma H, van Schayck OCP, Joore MA: Lifetime cost-effectiveness and equity impacts of the Healthy Primary School of the Future initiative. BMC Public Health 2020, 20(1):1887.

21. Martos-Moreno GA, Martinez-Villanueva J, Gonzalez-Leal R, Barrios V, Sirvent S, Hawkins F, Chowen JA, Argente J: Ethnicity Strongly Influences Body Fat Distribution Determining Serum Adipokine Profile and Metabolic Derangement in Childhood Obesity. Front Pediatr 2020, 8:551103.

22. World Health Organization (WHO). Growth reference data for 5-19 years: WHO [https://www.who.int/tools/growth-reference-data-for-5to19-years/indicators/bmi-for-age]

23. de Onis M, Lobstein T: Defining obesity risk status in the general childhood population: which cut-offs should we use? Int J Pediatr Obes 2010, 5(6):458-460.

24. Rotevatn TA, Overgaard C, Melendez-Torres GJ, Mortensen RN, Ullits LR, Hostgaard AMB, Torp-Pedersen C, Boggild H: Infancy weight gain, parental socioeconomic position, and childhood overweight and obesity: a Danish register-based cohort study. BMC Public Health 2019, 19(1):1209.

. 

- At lines 224-225 the sentence "A reduction of at least 0.21~0.88 kg/m2 of BMI (only 0.86%~3.59% of the baseline BMI) in children with GOB..." is not clear to me; specifically, what do you mean with "(only 0.86%~3.59% of the baseline BMI)"?

Response: We apologize for this and have clarified this sentence. This sentence means that a reduction of 0.21-0.88 kg/m2, and the average baseline BMI of children with general obesity was 24.49kg/m2, so 0.86%-3.59% (0.21/24.49=0.86%, and 0.88/24.49=3.59%) of the baseline BMI will beneficial to the BP profile. We have clarified it in the text as follows (Line 277-279).

A reduction of at least 0.21~0.88 kg/m2 of BMI (only 0.86%~3.59% of the baseline BMI, baseline BMI=24.49 kg/m2, 0.21/24.49=0.86%, and 0.88/24.49=3.59%) in children with GOB (baseline BMI=24.49 kg/m2) could be significantly beneficial to the BP profile compared with the reference group (Table 4). 

- At lines 274-276, the meaning of the sentence "those converted from obese to non-obese doubles the HBP risk compared with the persistent non-obese children" is clear enough, but I believe it should be re-phrased.

Response: Thanks for the comment. We have rephrased the sentence as follows (Line 344-345):

Our study found that compared with the persistent non-obese children, children who converted from obese to non-obese have a doubled risk of HBP.

- There is some other minor revision of English to be made; in the attached file I highlighted the line numbers where some correction is needed, IMHO. There are possibly more than those ones, but they are the ones I easily spotted.

Response: Many thanks for all the helpful editing, which could help us to improve the manuscript. According to your comments, we have revised our English throughout the manuscript.

Reviewer #3: 1. Did the Authors use some exclusion criteria to enroll the subjects?

Response: Thanks for the comment. There were inclusion and exclusion criteria to enroll the subjects. We have added the relative inclusion and excluding criteria in the manuscript (Line 104-107).

For the enrollment of subjects, the inclusion criteria were as follows: grade of 1 to 5 in primary schools and grade of 6 to 8 and 10 to 11 in secondary schools. And the exclusion criteria were as follows: Suffering or any history of disabilities, cardiovascular diseases, asthma, etc.

2. All the investigated variables were normally distributed? Which kind of test the Authors used to verify data normality?

Response: Thanks for the comment. We did Kolmogorov-Smirnov test to investigated variables’ normal distribution, and found that all investigated variables were not normally distributed. We have revised the description of the general characteristics of the variables as follows. And the description of analysis in the method section was also revised accordingly.

3. It would be better use BMI-SDS instead of BMI considering the age of subjects.

Response: Thanks for the helpful and professional comment. Age- and sex-specific BMI standard deviation score (BMI SDS) was calculated using the World Health Organization (WHO) child growth charts [1,2]. Then for the obese children, we categorized the then into 5 quintiles with P20, P40, P60, and P80 of BMI SDS change from baseline to the follow-up, similar results were observed. When compared with the reference group (quintile 5), quintile 1 group had significantly low risk of HBP (OR=0.68, 95%CI: 0.59-0.79). For quintile 2 group, OR was also <1 (tending to be protective, OR=0.87), but it didn’t reach statistical significance. We have added this as a part of our sensitivity analysis as follows. 

In the method section (Line 187-190)

(5) We also used BMI standard deviation score (BMI SDS) instead of BMI to compare the risk of HBP in different groups with multivariate logistic models. Age- and sex-specific BMI SDS was calculated using the World Health Organization (WHO) child growth charts[25-27].

In the result section (Line 282-284)

In addition, BMI SDS was used instead of BMI to test the association of different quintiles of BMI SDS change and risk of HBP, similar results were observed (S8 Table).

25. McLaughlin EJ, Hiscock RJ, Robinson AJ, Hui L, Tong S, Dane KM, Middleton AL, Walker SP, MacDonald TM: Appropriate-for-gestational-age infants who exhibit reduced antenatal growth velocity display postnatal catch-up growth. PLoS One 2020, 15(9):e0238700.

26. Zhang L, Huang L, Zhao Z, Ding R, Liu H, Qu W, Jia X: Associations Between Delivery Mode and Early Childhood Body Mass Index Z-Score Trajectories: A Retrospective Analysis of 2,685 Children From Mothers Aged 18 to 35 Years at Delivery. Front Pediatr 2020, 8:598016.

27. Poh BK, Lee ST, Yeo GS, Tang KC, Noor Afifah AR, Siti Hanisa A, Parikh P, Wong JE, Ng ALO, Group SS: Low socioeconomic status and severe obesity are linked to poor cognitive performance in Malaysian children. BMC Public Health 2019, 19(Suppl 4):541.

4. Why the Authors decided to categorize the adiposity parameter (BMI or WHtR) change during the follow-up as quintiles?

Response: We thank the reviewer for this comment. We categorized (BMI or WHtR change) as quintiles [1] because we want to find the potential BMI or WhtR reduction range of obese children to get an improved BP profile. We further categorized the reduction range of obese children into ten equal groups with similar results observed.

[1] Carthy P, Lyons S, Nolan A. Characterising urban green space density and footpath-accessibility in models of BMI[J]. BMC Public Health, 2020, 20(1).

5. In logistic regression, in addition to OR they have to show and mention also the goodness of fit test of the model. To adjust a regression for multiple variables without to verify if these variables are possible confounders is not correct. Mantel–Haenszel method could be used to verify if a variable is a possible confounder.

Response: Thanks for this comment. For logistic regression models exploring the association between risk of high blood pressure and different general or abdominal obesity status change, from model 1 to model 4, the goodness of fit test improved gradually, the Nagelkerke R2 of model 1 and model 4 for general obesity change pattern were 5.9% and 13.4%, and 4.8% and 13.2% for model 1 & 4 for abdominal obesity change patterns. For the categorical covariates in the logistic regression models, we used Mantel–Haenszel method to verify whether sex, age group(<=13 years old and >13 years old), and area (rural and urban) are potential cofounders, the results showed that they were possible confounders which were also found in previous studies[1-3]. These variables were adjusted when the risk factors for high blood pressure were analyzed.

[1] Yang YD, Song J Y, Wang S, et al. Combined effects of the rs9810888 polymorphism in calcium voltage-gated channel subunit alpha1 D (CACNA1D) and lifestyle behaviors on blood pressure level among Chinese children[J]. PLoS ONE, 2019, 14(5):e0216950-.

[2] Yang Y, Dong B , Wang S , et al. Prevalence of high blood pressure subtypes and its associations with BMI in Chinese children: a national cross-sectional survey[J]. BMC Public Health, 2017, 17(1):598.

[3] Yang Y, Dong B , Zou Z , et al. Association between Vegetable Consumption and Blood Pressure, Stratified by BMI, among Chinese Adolescents Aged 13–17 Years: A National Cross-Sectional Study[J]. Nutrients, 2018, 10(4):451.

6. Could the Authors clarify the sentence a” reduction of at least 0.21-~0.88 kg of BMI (only 0.86%~3.59% of the baseline BMI) in children with GOB (baseline BMI=24.49 kg/m2) at lines 224. It is not very clear: only 0.86%~3.59% of the baseline BMI.

Response: We apologize for this and have clarified this sentence. This sentence means that a reduction of 0.21-0.88 kg/m2, and the average baseline BMI of children with general obesity was 24.49kg/m2, so 0.86%-3.59% (0.21/24.49=0.86%, and 0.88/24.49=3.59%) of the baseline BMI will beneficial to the BP profile. We have clarified it in the text as follows (Line 277-279).

A reduction of at least 0.21~0.88 kg/m2 of BMI (only 0.86%~3.59% of the baseline BMI, baseline BMI=24.49 kg/m2, 0.21/24.49=0.86%, and 0.88/24.49=3.59%) in children with GOB (baseline BMI=24.49 kg/m2) could be significantly beneficial to the BP profile compared with the reference group (Table 4).

---

## [Editor Report · Decision Letter 1]

25 Aug 2021

Impact of short-term change of adiposity on risk of high blood pressure in children: Results from a follow-up study in China.

PONE-D-21-04822R1

Dear Dr. Yang,

We’re pleased to inform you that your manuscript has been judged scientifically suitable for publication and will be formally accepted for publication once it meets all outstanding technical requirements.

Kind regards,

Raffaella Buzzetti, M.D.

Academic Editor

PLOS ONE

---

## [Editor Report · Acceptance letter]

1 Sep 2021

PONE-D-21-04822R1 

Impact of short-term change of adiposity on risk of high blood pressure in children: Results from a follow-up study in China. 

Dear Dr. Yang:

I'm pleased to inform you that your manuscript has been deemed suitable for publication in PLOS ONE. Congratulations! Your manuscript is now with our production department. 

Kind regards, 

on behalf of

Dr. Raffaella Buzzetti 

Academic Editor

PLOS ONE